# Alterations in the Structure, Composition, and Organization of Galactosaminoglycan-Containing Proteoglycans and Collagen Correspond to the Progressive Stages of Dupuytren’s Disease

**DOI:** 10.3390/ijms25137192

**Published:** 2024-06-29

**Authors:** Luiz Guilherme S. Lenzi, João Baptista Gomes dos Santos, Renan P. Cavalheiro, Aline Mendes, Elsa Y. Kobayashi, Helena B. Nader, Flavio Faloppa

**Affiliations:** 1Department of Orthopaedics and Traumatology, Escola Paulista de Medicina, Universidade Federal de São Paulo, São Paulo 04038-032, SP, Brazil; lenzilgs@gmail.com (L.G.S.L.); joao.epm@hotmail.com (J.B.G.d.S.); faloppa.dot@unifesp.br (F.F.); 2Molecular Biology Program, Instituto de Farmacologia e Biologia Molecular, Escola Paulista de Medicina, Universidade Federal de São Paulo, São Paulo 04024-002, SP, Brazil; rpcavalheiro@gmail.com (R.P.C.); alina.mendess@gmail.com (A.M.); elsayoko@gmail.com (E.Y.K.); 3Faculdade de Medicina ABC, Centro Universitário, Santo André 09060-870, SP, Brazil

**Keywords:** sulfated glycosaminoglycans, chondroitin and dermatan sulfates, hyaluronic acid, proteoglycans, collagen, fascia palmar, extracellular matrix, versican, decorin

## Abstract

Dupuytren’s disease (DD) is a prevalent fibroproliferative disorder of the hand, shaped by genetic, epigenetic, and environmental influences. The extracellular matrix (ECM) is a complex assembly of diverse macromolecules. Alterations in the ECM’s content, structure and organization can impact both normal physiological functions and pathological conditions. This study explored the content and organization of glycosaminoglycans, proteoglycans, and collagen in the ECM of patients at various stages of DD, assessing their potential as prognostic indicators. This research reveals, for the first time, relevant changes in the complexity of chondroitin/dermatan sulfate structures, specifically an increase of disaccharides containing iduronic acid residues covalently linked to either N-acetylgalactosamine 6-*O*-sulfated or N-acetylgalactosamine 4-*O*-sulfated, correlating with the disease’s severity. Additionally, we noted an increase in versican expression, a high molecular weight proteoglycan, across stages I to IV, while decorin, a small leucine-rich proteoglycan, significantly diminishes as DD progresses, both confirmed by mRNA analysis and protein detection via confocal microscopy. Coherent anti-Stokes Raman scattering (CARS) microscopy further demonstrated that collagen fibril architecture in DD varies importantly with disease stages. Moreover, the urinary excretion of both hyaluronic and sulfated glycosaminoglycans markedly decreased among DD patients.Our findings indicate that specific proteoglycans with galactosaminoglycan chains and collagen arrangements could serve as biomarkers for DD progression. The reduction in glycosaminoglycan excretion suggests a systemic manifestation of the disease.

## 1. Introduction

Dupuytren’s disease (DD) is a common fibroproliferative condition of the hand influenced by genetic, epigenetic, and environmental factors [1]. It exhibits a global prevalence of approximately 8%, varying by region: 17% in Africa, 15% in Asia, 10% in Europe, and 2% in the Americas [2]. The disease leads to the formation of knots and fibers in the palmar fascia, characterized by the flexion contractures of fingers, and involves a reduction in the production of types I and III collagens as well as metalloproteinases (MMP-2, MMP-9, MMP-13) based on mRNA assessments in myofibroblast cultures from patients [3]. Conversely, Johnston et al. (2006) identified an increase in key collagenases, namely MMP1, MMP13, and MMP14 [4,5,6].

Myofibroblasts, the primary cells involved in DD, can contract and exhibit characteristics of both fibroblasts and smooth muscle cells, alongside high levels of proinflammatory cytokines in tissues from affected patients [7]. Growth factors such as TGF-beta, TNF-alpha, PDGF, and GM-CSF have been implicated in the abnormal cell activation, collagen synthesis, and extracellular deposition associated with the disease [8,9,10].

Glycosaminoglycans, critical linear anionic polysaccharides in various biological processes, exist mostly as proteoglycans in tissues, with polysaccharide chains covalently linked to a core protein, except for hyaluronic acid. These molecules are composed of alternating units of hexosamine and uronic acid, connected by glycosidic linkages that vary in type and position. The type of hexosamine (D-glucosamine or D-galactosamine), the type of uronic acid (D-glucuronic acid or L-iduronic acid), the type (α or β) and the relative position (1→3 or 1→4) of the glycosidic linkages distinguish the different glycosaminoglycans. Sulfation occurs at different positions of the disaccharide units and embrace heparan sulfate (HS), heparin (Hep), chondroitin sulfate (CS) and dermatan sulfate (DS) [11,12,13,14,15].

The extracellular matrix (ECM) is a complex network comprising collagens, fibronectins, laminins, proteoglycans, and hyaluronic acid. Alterations in the ECM’s composition and structure can significantly impact both normal physiological functions and pathological conditions [16,17]. Dupuytren disease seems to result from a combination of heritable genomic factors and abnormal environmental stimuli. Despite ongoing research, there is currently no cure for DD, and treatments remain palliative [18,19,20,21,22,23,24,25,26,27,28].

The extracellular matrix of the palmar fascia is composed of fibrillar and non-fibrillar components. Among the fibrillar components are some microfibrillar components such as collagen fibrils, and elastic fibers. The non-fibrillar components are represented by proteoglycans and non-collagenous glycoproteins. The collagen fibers are elongated, birefringent, non-elastic, and non-branching, although bundles of collagen fibrils may leave one fiber and connect to another, giving the impression of branching. Collagen is a glycoprotein that constitutes the largest class of insoluble fibrous proteins in the ECM. Its fibers exhibit the morphological characteristic of axial periodicity, produced by the overlapping of tropocollagen molecules that form fibrils arranged in bundles or fascicles parallel to the longitudinal axis of the tendon. Connective tissue cells occur between the collagen bundles. Most of these cells correspond to fibroblasts, whose function is related to the synthesis of fibers, glycoproteins, and proteoglycans of the matrix. The palmar fascia is mainly rich in type I collagen, whereas types II, III, and V collagens are also present. Type I collagen is the most common, forming coarse fibers in connective tissue such as normal skin pattern, fascia, tendon, bone, and dentin. Type II collagen forms thin fibers and is almost exclusively present in the matrices of hyaline and elastic cartilage. Type III collagen, often associated with type I, is also known as thinner reticular fiber. Type V collagen is characterized by fibrils that bind to type I and type III collagens, forming heterotypic fibrils, thus having a broad distribution similar to that of type I collagen [29,30,31].

Our preliminary studies indicated an increase in dermatan sulfate in DD patients’ extracellular environments [9]. The present research aims to further examine the contents and organization of glycosaminoglycans, proteoglycans, and collagen in the ECM of patients at various stages of Dupuytren’s contracture as a potential prognostic factor for the disease. We observed notable changes in the content and structure of chondroitin and dermatan sulfates, as well as of the high molecular weight proteoglycan (versican), small leucine-rich proteoglycan (decorin), and collagen arrangement in the palmar fascia of patients compared to normal tissue, correlating with disease progression. Additionally, urine analyses from DD patients have shown a significant decrease in the excretion of hyaluronic acid and sulfated glycosaminoglycans.

## 2. Results

### 2.1. Distribution of Glycosaminoglycans in DD’s Palmar Fascia

Initially, the excised palmar fascia specimens from 14 patients were segmented into three regions (P, M, D) according to previously indicated stages of the disease (I–IV). All patients exhibited increased levels of sulfated glycosaminoglycans (heparan sulfate, dermatan sulfate, and chondroitin sulfate) as well as of hyaluronic acid compared to controls. Nevertheless, no statistical differences were observed when comparing the three regions P, M, and D for each individual stage of the disease (Figure 1). This result could mean that the variations observed in these regions are not enough to be considered, suggesting that these regions behave similarly or have similar properties in the context of the present study.

Consequently, for subsequent experiments, the three regions were treated as distinct measurements for the same patient (Figure 2). Significant increase in the contents of chondroitin sulfate (CS) (Figure 2A), dermatan sulfate (DS) (Figure 2B) and heparan sulfate (Figure 2C) is observed for each stage when compared to control palmar tissue. The results also show that DS is the main SGAG present in DD palmar tissue. Also, the data indicates a decrease in the amount of CS with the progression of the disease, from stages I and II to stages III and IV (Mann-Whitney, *p* < 0.05). Furthermore, the amounts of hyaluronic acid (Figure 2D) also increase when compared to control tissue. Also, HA is present in much lower concentrations compared to SGAG.

### 2.2. Structural Characteristics of Sulfated Galactosaminoglycans in DD’s Palmar Fascia

Some details of the structure of the sulfated galactosaminoglycans, that is, chondroitin and dermatan sulfates were revealed by degradation with chondroitinases and identification of the unsaturated disaccharides. Based on the specificity of chondroitinase AC one can conclude that there is a decrease in the percentage of disaccharides containing beta-D-glucuronic acid and beta-D-N-acetylgalactosamine 6-*O* sulfated, according to the stage of the disease [11]. On the other hand, chondroitinase ABC recognizes the N-acetylgalctosamine linked to either beta-D-glucuronic or alpha-L-iduronic acid residue. Thus, the degradation products clearly indicate an increase in disaccharides containing iduronic acid residues which are prevalent in dermatan sulfate chains. Again, a decrease in disaccharides bearing 6-*O*-sulfation is observed. It is relevant to point out that the DS like material present in the DD’s palmar fascia contains iduronic acid linked to either N-acetylgalactosamine 6-*O*-sulfated or N-acetylgalactosamine 4-*O* sulfated (Table 1). The fine structure of the CD/DS from DD’s palmar fascia clearly shows a decrease in disaccharides 6-*O*-sulfated containing glucuronic acid which parallels the severity of the disease. Also, an increase in disaccharides containing iduronic acid linked to N-acetylgalactosamine 6-*O* sulfated or 4-*O* sulfated occurs with the gravity of the disease. The combined data clearly indicates most relevant changes in the fine structure of chondroitin sulfate and dermatan sulfate in patients with DD. There is an increase in disaccharides bearing alpha-L-iduronic acid residues linked to either beta-D-N-acetylgalactosamine 4 or 6-*O*-sulfated. This results are in agreement with data obtained by Alfonso-Rodríguez et al. (2014) by microarray analysis without validation of mRNA for epimerase and sulfotransferases [32].

### 2.3. Proteoglycans and Colagens in DD’s Palmar Fascia

These results prompted further investigation into the expression of proteoglycans (PGs) containing CS or DS chains covalently linked to the core protein. Figure 3 presents the qPCR results where the three different regions are included as samples from the same patient. Two high molecular weight PGs (HMWPGs) containing CS/DS chains were examined, revealing a significant increase in their expression from stages I to IV (Figure 3A,B). However, versican expression was approximately eight to ten times higher than that of aggrecan. Among the small leucine-rich proteoglycans (SLRPs), the expression of decorin and biglycan was investigated. Significant increases in mRNA for decorin were observed only in stages I and II (Figure 3C), with no changes in biglycan expression (Figure 3D).

The expression levels of versican and decorin, as investigated by confocal microscopy, are consistent with the qPCR data (Figure 4). Versican expression increases with the severity of the disease (stages I to IV), and the images suggest changes in the distribution and arrangement of this PG in the ECM. Interestingly, decorin expression decreases from stages I to IV, corroborating the qPCR results.

Hyaluronic acid (HA) expression, analyzed by immunohistochemistry (Figure 5), parallels the direct measurement results of HA in the samples (Figure 2D).

Collagen architecture, including fibers and bundles, was investigated using second harmonic generation in CARS microscopy, clearly showing alterations in collagen fiber orientation from control to DD samples (Figure 6A). The angular distribution within the collagen samples can be quantified (Figure 6B), revealing three major peaks (−45, 45, and zero degrees) in DD stages, whereas control orientations show two peaks at −90 and 90 degrees, highlighting disorganization in collagen arrangement in the diseased tissues compared to control.

### 2.4. Urinary Glycosaminoglycans from DD Patients

The contents of SGAG (chondroitin sulfate and heparan sulfate) in urine from controls and patients with different stages of DD is shown in Figure 6A. It is observed a significant decrease for all four compared to control. Controls show a mean value of 16.0 μg SGAG/mg creatinine whereas for the 4 different stages of the disease the values vary between 5.6 to 9.7 μg SGAG/mg creatine. The distribution and mean values of HA in urine of controls and DD patients can be depicted in Figure 7C,D. Controls show a mean value of 24.6 ng HA/μg creatinine whereas for the 4 different stages of the disease the values vary between 6.8 to 10.7 (ng HA/μg creatinine).

The excretion of sulfated glycosaminoglycans and hyaluronic acid do not vary among the different stages of the disease but are significantly decrease compared to normal subjects.

## 3. Discussion

The combined results demonstrate alterations in the ECM composition and architecture in patients with Dupuytren’s disease (DD). Significant changes in sulfated glycosaminoglycans (SGAGs), particularly increases in dermatan sulfate and chondroitin sulfate, were noted. Additionally, an elevation in hyaluronic acid levels was observed across all stages of the disease. Our investigations into the fine structure of the chondroitin/dermatan sulfate chains revealed an increase in disaccharides containing iduronic acid covalently linked to N-acetylgalactosamine 6-*O* sulfated or N-acetylgalactosamine 4-*O* sulfated, correlating with increased DD severity. This finding is reported for the first time in the literature.

Furthermore, significant alterations in both decorin and versican within the extracellular matrix of DD patients were identified. Notably, versican expression, a high molecular weight galactosaminoglycan-containing proteoglycan, increased with advancing disease stages, while decorin, a small leucine-rich dermatan sulfate proteoglycan, decreased from stages I to IV, as detected by mRNA analyses and confocal microscopy. Coherent Anti-Stokes Raman Scattering (CARS) microscopy showed that collagen fibril architecture in DD is disrupted, becoming increasingly disorganized as the disease progresses. Decorin, known for its role in collagen fibril assembly, is linked to these structural changes [33,34,35,36,37,38].

Interestingly, studies conducted using palmar fascia from the lesion of DD and from clinically unaffected by Dupuytren as well as from normal forehand fascia, using different methodological approaches showed that the immunohistochemical expression of decorin was high in DD, whereas versican was highly expressed in the normal fascia, with no differences for aggrecan, besides increase in collagen content. Nevertheless, in that paper the degree of disease severity was not taken into consideration. Also, we want to point that another interesting aspect of the paper was the finding that the clinically unaffected tissue showed changes in some ECM components, indicating that it may be affected. These findings underscore the subtle yet significant changes occurring in the ECM even in clinically unaffected tissues, highlighting the systemic nature of the disease [32]. Furthermore, the authors evaluated the expression of ECM-related genes by microarray using the information provided by Affymetrix and they found dermatan sulfate epimerase, sulfotransferases and hyaluronic acid synthases increase in the DD samples as revealed in our study by the analysis of the structural characteristics of the chondroitin/dermatan sulfate structure. 

Additionally, our study confirms previous findings that versican not only binds directly to collagen, regulating its structure and mechanics but also that its organization within the ECM depends on interactions with other components like hyaluronic acid.

The combined data shows that the architecture of collagen fibrils in DD is most relevant for the symptoms that lead to the contracture, as observed for capsule in frozen shoulders patients [39].

Collectively, these data suggest that the architecture of collagen fibrils in DD is crucial for the symptoms leading to contracture, akin to what is observed in patients with frozen shoulder syndrome. The decreased excretion of hyaluronic acid and sulfated glycosaminoglycans in the urine of DD patients further suggests a systemic impact of the disease. These findings underscore the critical roles of glycosaminoglycans and proteoglycans in the formation and assembly of interstitial collagen fibers within the palmar fascia of DD patients, influencing the overall ECM structure and contributing to the disease’s progression and symptoms.

The research also found decreased excretion of hyaluronic acid and sulfated glycosaminoglycans in the urine of DD patients, suggesting a systemic impact of the disease. Moreover, the findings support the crucial roles of glycosaminoglycans and proteoglycans in forming interstitial collagen fibers and assembling fibrils into fibers within the palmar fascia of DD patients, thereby influencing the overall structure of the ECM.

## 4. Material and Methods

### 4.1. Surgical Specimens

This study, conducted at the Department of Orthopedics and Traumatology, Escola Paulista de Medicina, Universidade Federal de São Paulo, involved collecting palmar fascia samples from patients diagnosed with Dupuytren’s Disease (DD). These samples were categorized based on contracture severity (stages I, II, III, IV). Ethical approval was granted by the appropriate committee (Approval Nr. 48734315.2.0000.5505), and all participants provided informed consent. Inclusion criteria were patients with DD indicated for surgery who had not undergone previous surgical interventions or experienced significant trauma. Post-surgical follow-up occurred at intervals of 7, 14, 30, 60, and 90 days to monitor surgical outcomes. Disease staging was determined using cumulative goniometry deficits, measured at the dorsal aspect of the affected fingers, following the method described by Tubiana et al. (1981) [40]. The number of patients in each stage were as follows: I (4), II (4), III (2), IV (4). Regarding the categorization of the disease, the stages of the severity of the contracture were guided by the Tubiana Classification, which is based on the sum of the extension deficits of the affected fingers, that is, the flexion deformity of each affected finger. The sum of the extension deficit of each joint (metacarpophalangeal, proximal interphalangeal, and distal interphalangeal) results in the degree of flexion deformity. When there is hyperextension of the distal interphalangeal joint, the degree of hyperextension is added to the total flexion deformity of the other joints. In stage I, the result of the sum of the extension deficit is between 0° and 45°; in stage II, the sum of the extension deficit is between 45° and 90°; in stage III, it is between 90° and 135°; and in stage IV, the sum exceeds 135°. The patient was categorized according to the finger with the highest degree of involvement by the disease [41].

In the present study, palmar fascia samples extracted from patients undergoing surgery for carpal tunnel syndrome served as controls. We selected these individuals because carpal tunnel syndrome does not involve pathological changes in the palmar fascia, making their fascial tissue ideal for comparison. The samples were conveniently obtained during routine surgical procedures for neurological decompression, where access to unaffected palmar fascia is straightforward. This approach ensures that the collection of control samples does not interfere with the surgical technique or compromise patient safety. The primary objective of using fascia from control patients is to provide a baseline for comparing the pathological features observed in patients with diseases specifically affecting the palmar fascia.

### 4.2. Surgical Procedure

Partial fasciectomies were performed using a standardized technique, including plexus block anesthesia and limb exsanguination using an Esmarch band. Rigorous hemostasis was achieved, and wounds were closed using sterile nylon sutures (4-0 or 5-0). The fibrotic cords, approximately 2.5 cm in length, were excised and segmented into proximal (P), median (M), and distal (D) regions for analysis [42]. Tissue samples were preserved in acetone at room temperature for glycosaminoglycan (GAG) studies, in TRIzol at −80 °C for PCR analyses, and in 4% paraformaldehyde at 4 °C for histological examination. Additionally, 50 mL of urine was collected from each patient pre-operatively and stored in the presence of thymol crystals at 4 °C for subsequent analysis.

### 4.3. Glycosaminoglycan Analysis in Palmar Aponeurosis Samples

The palmar aponeurosis samples were washed in phosphate-buffered saline (PBS) at pH 7.4, ground in acetone (10 vol), kept overnight at room temperature, and the samples colleted by centrifugation (3000× *g* for 15 min), dried, weighed, and proteolyzed with maxatase (Biocon Laboratories, Brazil) at a concentration of 4 mg/mL in 0.05 M phosphate buffer, pH 7.0, for 18 h at 60 °C, as described [43,44]. Subsequently, peptides and nucleic acid fragments were removed by precipitation with trichloroacetic acid at a final concentration of 10%, at 4 °C. Following another centrifugation (10 min at 3500× *g*, 4 °C), the supernatant containing glycosaminoglycans (GAG) was precipitated by adding methanol (3 vol, −20 °C, 18 h). The resulting precipitate was collected by centrifugation and dried. The samples were reconstituted in distilled water and analyzed in triplicate for sulfated glycosaminoglycans (SGAG) and hyaluronic acid (HA) content. SGAG levels were identified and quantified using agarose gel electrophoresis in 0.05 M 1,3-diaminopropane acetate (PDA) buffer, pH 9.0, employing standards of chondroitin sulfate (CS), dermatan sulfate (DS), and heparan sulfate (HS) (Seikagaku, Japan). The quantitative results were expressed in micrograms of SGAG per milligram of dry tissue, with values reported as means along with their lower and upper limits of 95% confidence intervals. HA determination was performed using a highly specific enzyme-linked immunosorbent assay (ELISA)-like fluorometric method [45]. The absolute amounts of HA were expressed in nanograms per milligram of dry tissue.

### 4.4. Glycosaminoglycan Analysis in Urine Samples

The glycosaminoglycans in urine samples were extracted using gel filtration chromatography as previously described [46]. For this process, a 5 mL aliquot of urine was applied to a Sephadex G25 column equilibrated with distilled water. After discarding the inclusion volume, the subsequent 10 mL flow-through was collected, vacuum dried, and finally dissolved in 25 µL of distilled water. The samples were then stored at −20 °C for further analysis. This purification method effectively separates glycosaminoglycans (GAG) from salts and other impurities such as pigments. The sulfated glycosaminoglycans (SGAG) were identified by agarose gel electrophoresis on a PDA agarose gel. Creatinine levels were determined using the alkaline picrate method (Labtest Diagnostica SA, Lagoa Santa, MG, Brazil). The concentration of SGAG were expressed as µg SGAG per mg of creatinine. Additionally, the amounts of hyaluronic acid (HA) were determined using a non-competitive “ELISA-like” fluorometric method and expressed as ng HA per µg of creatinine [45].

### 4.5. Enzymatic Degradation with Chondroitinases AC and ABC

The samples (25 µg) were incubated with 0.1 U of chondroitinase AC (Flavobacterium heparinum, Sigma-Aldrich, St. Louis, MO, USA) in 0.02 M phosphate buffer and 0.1 U of chondroitinase ABC (*Proteus vulgaris*, Sigma-Aldrich) in 0.05 M Tris buffer (pH 8.0) at 37 °C for approximately 24 h. The analysis of disaccharides was performed using SAX-HPLC ion exchange chromatography on a SphereClone 80A column (Phenomenex Torrance, CA, USA), employing a 0–2 M NaCl gradient. Disaccharides were eluted at a final concentration of 1 M. The column had been previously calibrated with isolated disaccharides, and the elution profile was monitored at 230 nm.

### 4.6. RNA Extraction, Real-Time Reverse Transcription PCR and qPCR Analysis

Total RNA was isolated using TRIzol^®^ Reagent (Invitrogen), quantified spectrophotometrically, and employed as the template for the reverse transcription reaction. Complementary DNA (cDNA) was synthesized from 2 μg of total RNA using the Improm II™ Reverse Transcriptase kit (Promega, Madison, WI, USA). Quantitative RT-PCR amplification was conducted with a 2 μL aliquot of diluted cDNA, specific primers (Table 2), and the SYBR Green Master Mix (Applied Biosystems, Foster City, CA, USA). This amplification was performed on a 7500 Real-Time PCR System (Applied Biosystems, Foster City, CA, USA) with the following thermal profile: initial activation at 95 °C for 10 min, followed by 40 cycles of denaturation at 95 °C for 15 s, annealing at 61 °C for 1 min, and extension at 72 °C for 30 s. Relative gene expression was assessed using a standard dilution curve-based method for real-time qPCR, using the ΔΔCT approach. Results were reported in arbitrary units, with negative controls included to verify the absence of contamination. Expression values were normalized for each target gene against the housekeeping gene RPL 13A (ribosomal protein L13A) consistently expressed across different samples and conditions [47].

### 4.7. Immunofluorescence, Confocal Microscopy, and Immunohistochemistry

The palmar aponeurosis specimens were fixed in 4% formaldehyde in PBS, embedded in Tissue Tek at −20 °C, and sectioned into 10 µm slices with a cryostat, mounted on slides for immunofluorescence analysis, as previously described [48]. The slides were first incubated in 0.1M glycine in PBS for 10 min, then in a blocking buffer (5% FCS and 0.02% saponin in PBS) for 1 h at room temperature, and subsequently with primary polyclonal antibodies at 4 °C overnight. These primary antibodies (Santa Cruz Biotechnology, Santa Cruz, CA, USA) were diluted 1:200 and targeted decorin (sc-22613), biglycan (sc-33788), versican (sc-25831), and aggrecan (sc-25674). Following PBS wash, the samples were incubated with secondary antibodies conjugated with Alexa Fluor-594 or streptavidin conjugated with Alexa 594 at room temperature for 1.5 h. Nuclei staining was performed with DAPI (1:1000; Molecular Probes, Eugene, OR, USA) for 15 min at room temperature. The prepared slides were then mounted in Fluoromont G and analyzed using a Leica TCS SP8 laser scanning confocal microscope. Coverslips were affixed to glass microscope slides using Fluoromount-G (2:1 in PBS), and images were captured with a Plan-Apochromat × 63 objective (numerical aperture 1.4) under oil immersion. The images were displayed as maximum intensity projections from z-series confocal stacks. For HA detection using immunohistochemistry, the slides underwent a 30-min incubation in 10% hydrogen peroxide, washed and non-specific protein binding sites blocked with 5% FBS. The slides were subsequently incubated with biotinylated link protein at 4 °C overnight, washed, processed using the Universal Dako LSAB^®+^ kit (Dako, Glostrup, Hovedstaden, Denmark), counterstained with hematoxylin and eosin (H&E) and examined under an inverted light microscope (Axiovert 40 CFL; Zeiss, Ostalbkreis, Baden-Württemberg, Germany) at 40× magnification.

### 4.8. Coherent Anti-Stokes Raman Scattering (CARS) Microscopy

Specimens from patients at the different DD stages and control palmar tissue were analyzed using the TCS SP8 CARS Confocal Microscope, as previously described [48]. This system includes an inverted microscope (DMI 6000 CS Trino; Leica Microsystems GmbH, Wetzlar, Germany) equipped with a picoEmerald tunable light source (APE). The excitation light was focused through a plan apochromatic multi-immersion objective (HC PL APO CS2 20×/0.75). Epi-SHG (epi-second harmonic generation) was employed to detect second harmonic generation from collagen fibers. The images were displayed as maximum intensity projections, corresponding to the Z-series of confocal stacks, and were collected via tile scan and processed using Leica LAS AF software (Leica Microsystems GmbH, Wetzlar, Germany). Finally, the orientation analysis of the collagen structure was carried out using the second harmonic generation images of collagen fibers (Z-series of confocal stacks). This analysis was performed using the OrientationJ plugin of the ImageJ software (National Institutes of Health (NIH), Bethesda, Maryland, USA).

### 4.9. Statistical Analysis

All data collected were analyzed using appropriate statistical tests, which were selected based on the distribution types of the variables. The distributions were categorized as nonparametric for two unmatched groups, analyzed using GraphPad Prism 9. The values for each continuous variable were organized and described using the mean and standard deviation. To compare the means of two sample groups, the Student’s *t*-test followed by the Mann-Whitney test was employed. For studies involving more than two groups with nonparametric data, a one-way ANOVA followed by the Kruskal-Wallis test was used. A probability value of *p* ≤ 0.05 was considered statistically significant.

## Figures and Tables

**Figure 1 ijms-25-07192-f001:**
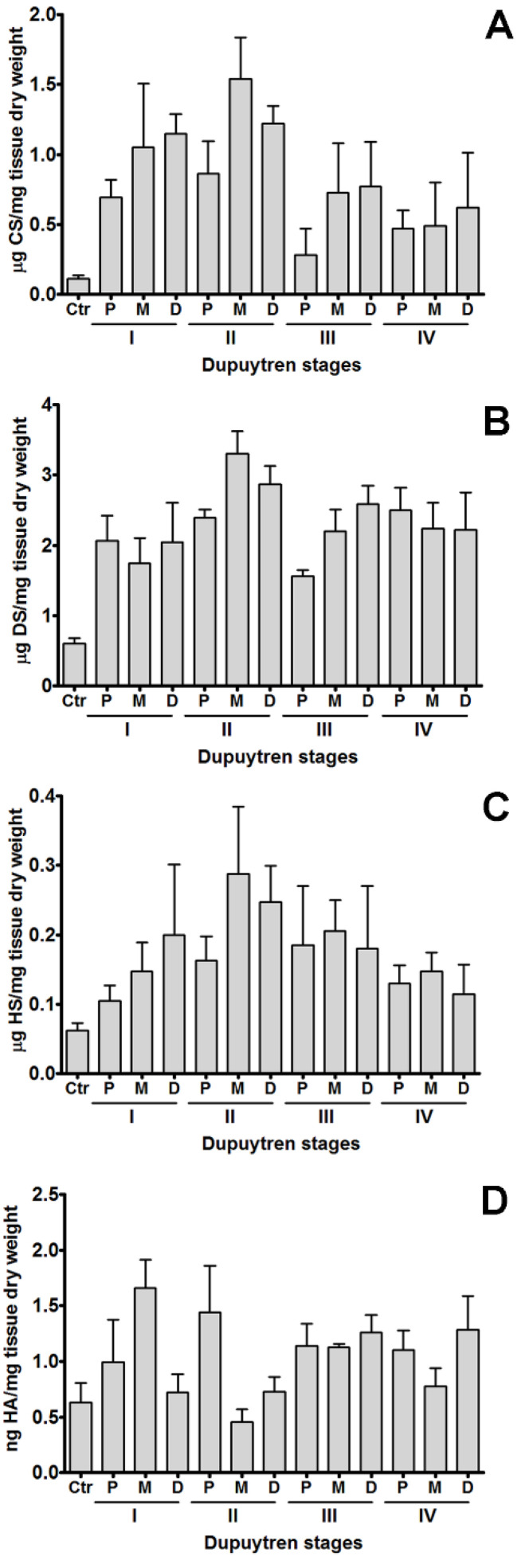
Chondroitin sulfate (**A**), dermatan sulfate (**B**), heparan sulfate (**C**) and hyaluronic acid (**D**) content in regions of palmar fascia from patients with different stages of Dupuytren’s disease. Each experiment was performed three times. Quantifications shown in the figure resume all experiments. No statistical significance was found for the contents of each individual glycosaminoglycan among the 3 regions of the palmar fascia for each stage of the disease. Error bars represent SD. The different stages of the disease show significant differences (*p* < 0.05).

**Figure 2 ijms-25-07192-f002:**
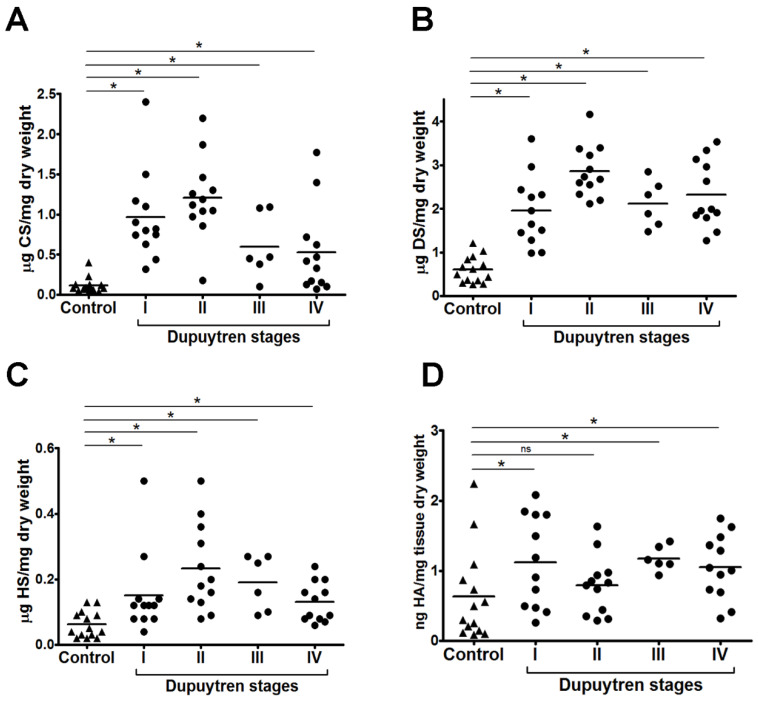
Chondroitin sulfate (**A**), dermatan sulfate (**B**), heparan sulfate (**C**) and hyaluronic acid (**D**) content in palmar fascia from patients with different stages of Dupuytren’s disease. Each experiment was performed three times. Quantifications shown in the figure resume all experiments. Hyaluronic acid (HA); chondroitin sulfate (CS); dermatan sulfate (DS); HS, heparan sulfate. The bar represents the mean. *, Mann-Whitney test, *p* < 0.05.

**Figure 3 ijms-25-07192-f003:**
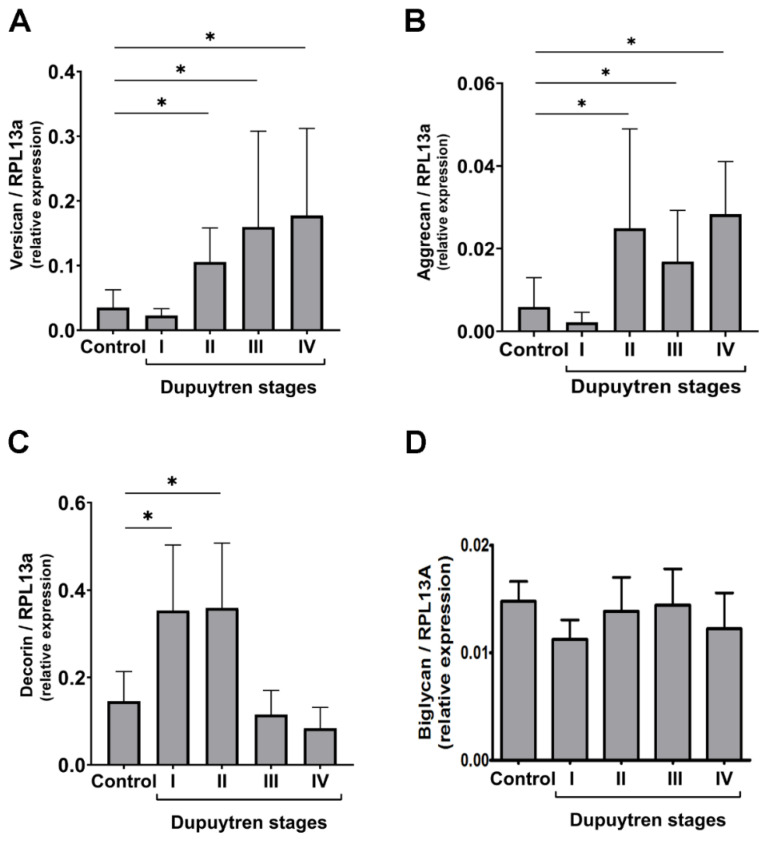
Expression of versican (**A**), aggrecan (**B**), decorin (**C**) and biglycan (**D**) in the palmar fascia from patients with different stages of Dupuytren’s disease. Each experiment was performed three times. Quantifications shown in the figure resume all experiments. The numbers indicate the expression of the studied gene relative to the housekeeping gene RPL 13A (ribosomal protein L13A). Error bars represent SD; *, Mann-Whitney test, *p* < 0.05. There is a significant difference when comparing stages I and II to stages III and IV (*p* < 0.05) for versican expression with an increase with the progression of the disease. On the other hand, there is a decrease in the expression of decorin when comparing stages I and II to stages III and IV (*p* < 0.05).

**Figure 4 ijms-25-07192-f004:**
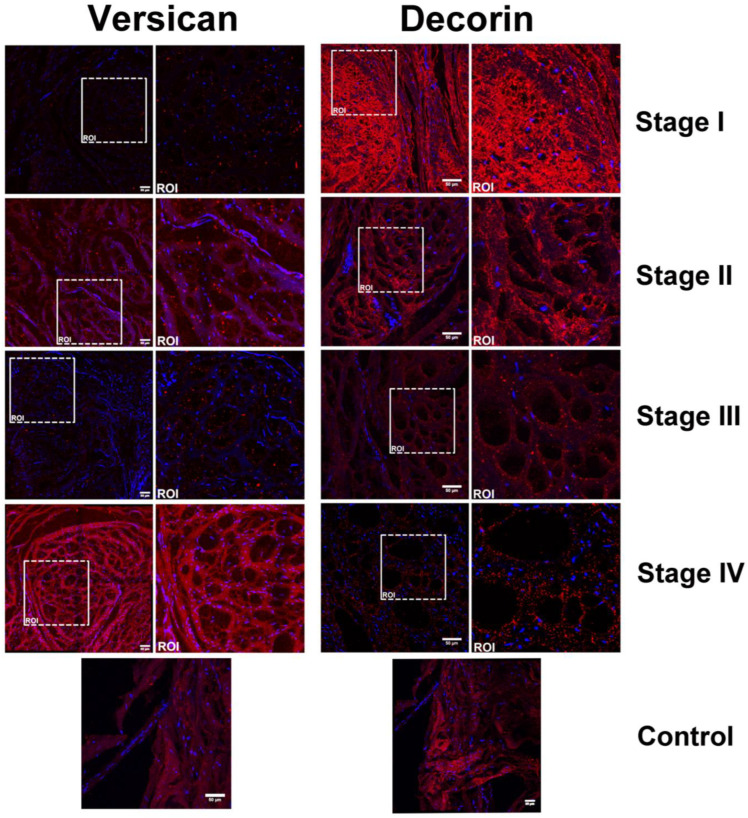
Expression of versican while decorin expressiondecreases with the rise in the severity of DD patients. The images represent confocal microscopy analyses of palmar fascia from patients with stages II and III of Dupuytren’s disease. Bars: 50 μm; ROI: region of interest. Images were captured with a Plan-Apochromat×63 objective (numerical aperture 1.4) under oil immersion.

**Figure 5 ijms-25-07192-f005:**
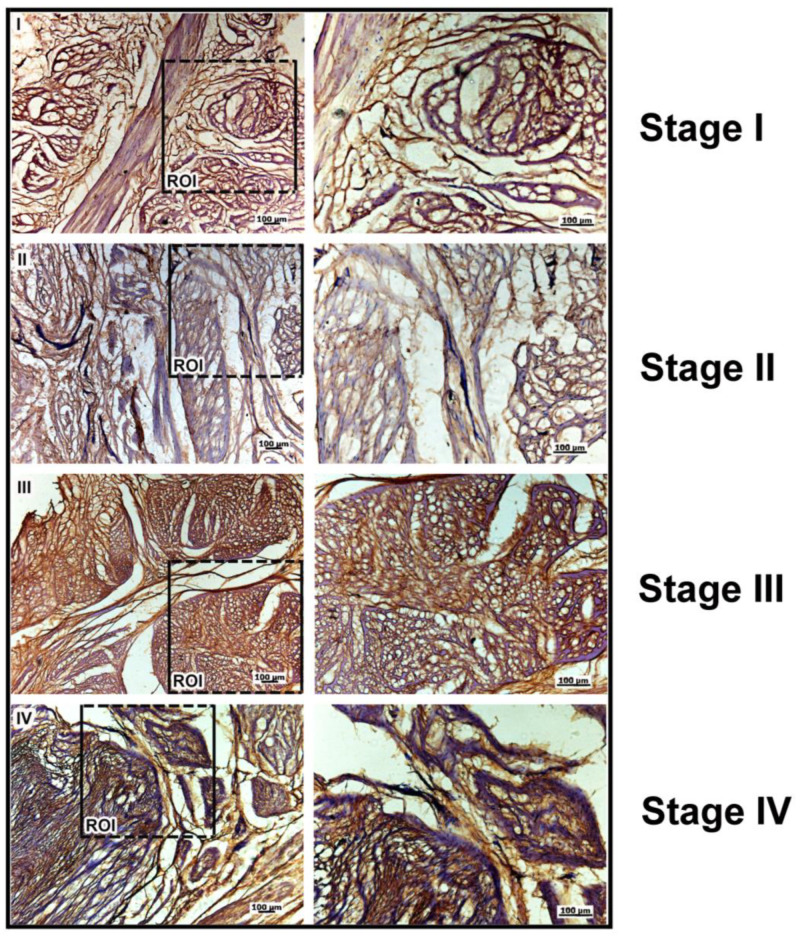
Immunohistochemistry for the detection of hyaluronic acid in different stages of Dupuytren’s disease. Bars: 100 μm; ROI: region of interest. The images were examined under an inverted light microscope (Axiovert 40 CFL, Zeiss) at 40× magnification.

**Figure 6 ijms-25-07192-f006:**
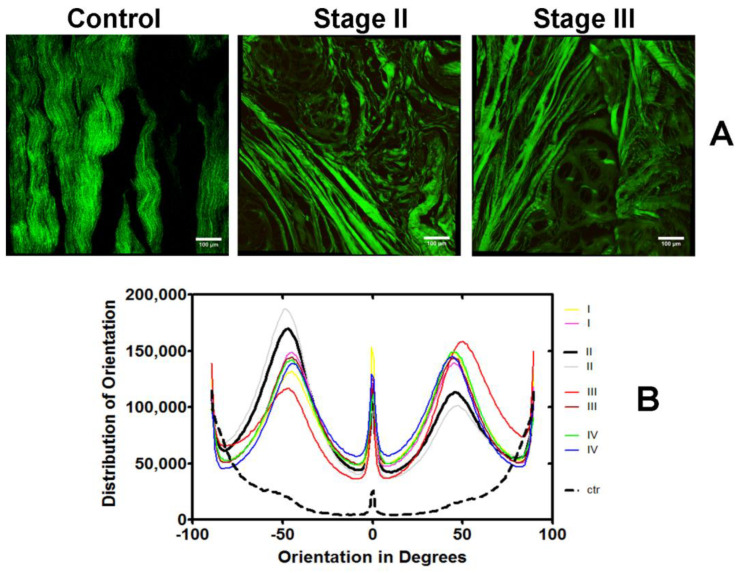
Coherent anti-stokes Raman scattering microscopy of collagen in Dupuytren’s disease. (**A**) The images represent scan of palmar sections for stages II and III of Dupuytren’s disease and control performed in both XY and Z planes. The projection shows in green the second harmonic generation signal of collagens. Bars: 100 μm. (**B**) Collagen orientation for stages I to IV of Dupuytren’s disease and control. The orientation analysis of collagen structure was performed using the second harmonic generation of collagen fibers images (Z-series of confocal stacks) from control and DD patients built by OrientationJ plugin of the ImageJ 1.52p software (National Institutes of Health (NIH), Bethesda, Maryland, USA). I–IV, different stages of Dupuytren’s disease; ctr, control.

**Figure 7 ijms-25-07192-f007:**
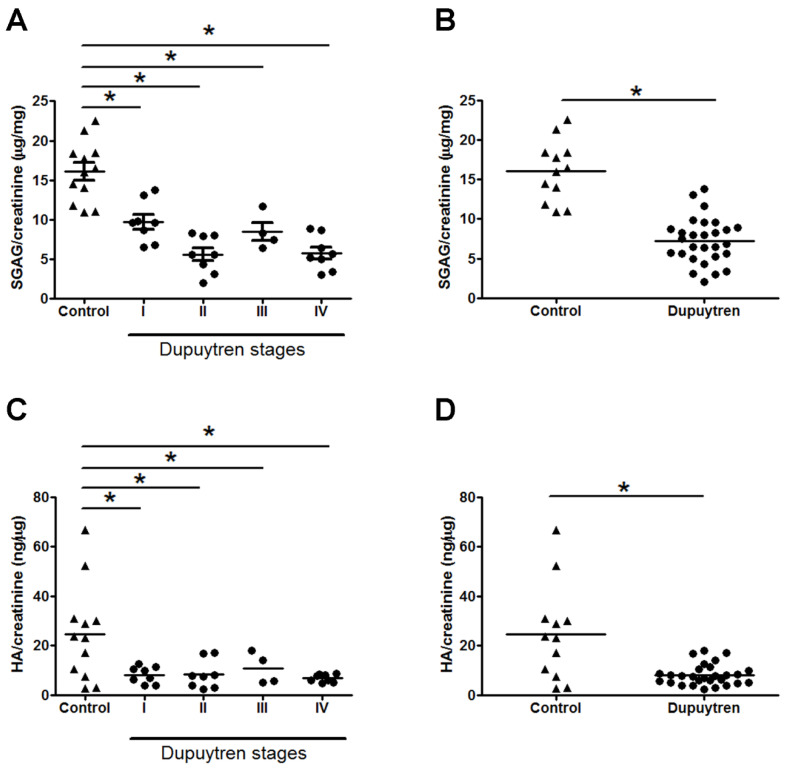
Urinary excretion of sulfated glycosaminoglycans (**A**) and hyaluronic acid (**C**) in patients with different stages of Dupuytren’s disease. Each experiment was performed three times. Quantifications shown in the figure resume all experiments. HA (**B**) and SGAG (**D**) urinary excretion in all patients independent of the stage of the disease. Each experiment was performed two times. The bar represents the mean. *, Mann-Whitney test, *p* < 0.05.

**Table 1 ijms-25-07192-t001:** Percentage of disaccharides from Dupuytren’s Disease stages after degradation with chondroitinases AC and ABC.

DD Stages	Chase AC (%)		Chase ABC (%)	
	GlcA-GalNAc6S	GlcA-GalNAc4S	GlcA-GalNAc6S + IdoA-GalNAc6S	GlcA-GalNAc4S + IdoA-GalNAc4S
I	84.9	15.1	66.1	33.9
II	83.6	16.4	19.3	80.8
III	77.2	22.8	30.7	69.3
IV	41.5	58.5	12.0	88.0
CS	51.1	48.9	47	53
DS	-	-	0	100

GlcA, glucuronic acid; IdoA, iduronic acid; GalNAc6S, N-acetylgalactosamine 6-*O* sulfated; GalNAc4S, N-acetylgalactosamine 4-*O* sulfated; I-IV, different stages of Dupuytren Disease; Chase AC and Chase ABC, respectively chondroitinases AC and ABC; CS, chondroitin sulfate; DS, dermatan sulfate.

**Table 2 ijms-25-07192-t002:** Primers used in Real Time PCR.

Gene	Forward (5′-3′)	Reverse (5′-3′)
Decorin	GCTTCTTATTCGGGTGTGAGT	TTCCGAGTTGAATGGCAGAG
Biglycan	CTCGTCCTGGTGAACAACAA	CAGGTGGTTCTTGGAGATGTAG
Versican	GTCACTCTAATCCCTGTCGTAATG	CTCGGTATCTTGCTCACAAAGT
Aggrecan	CCTTACCGTAAAGCCCATCTT	CCAGTCTCATTCTCAACCTCAG
RPL13a	TTGAGGACCTCTGTGTATTTGTCAA	CCTGGAGGAGAAGAGGAAAGAGA

## Data Availability

All data produced here are available upon request.

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
