# Peer review of "Alterations in the Structure, Composition, and Organization of Galactosaminoglycan-Containing Proteoglycans and Collagen Correspond to the Progressive Stages of Dupuytren’s Disease"

_ijms, 2024, doi:10.3390/ijms25137192_

Round 1

Reviewer 1 Report

Comments and Suggestions for Authors

The following points should be addressed:

   1. The authors should briefly describe the normal microanatomy of control and disease stages as it relates to cells and ECM. This will help clarfy the significance of the results.

   2. The authors use immunohistochemistry to help define expression of certain components. First this does not detect "expression" but rather "accumulation" and it is not necessarily quantitative. 

   3. Additional data such as quantitative chemical evaluation and/or western blots would strengthen this manuscript.

   4. Some discussion of the importance of the sulfate changes observed would seem appropriate. 

Author Response

We would like to thank the reviewers for their comments which certainly added to the manuscript in its present format. 

Reviewer # 1

The following points should be addressed:

  1. The authors should briefly describe the normal microanatomy of control and disease stages as it relates to cells and ECM. This will help clarify the significance of the results.

Answer: As requested by the reviewer we have included a description of the microanatomy in the manuscript in its present format.

  1. The authors use immunohistochemistry to help define expression of certain components. First this does not detect "expression" but rather "accumulation" and it is not necessarily quantitative.

Answer: we disagree with the reviewer that the immunohistochemistry does not relate to the expression of compounds. It is a routine method used for the evaluation of expression of compounds using different microscopic analyses. Immunohistochemistyry or immunocytochemistry detects the antigen of interest in tissue sections or cells and can be very useful for determining whether cells express a certain protein while also providing subcellular localization and can be used in multiplex format to determine whether multiple proteins are co-located. Furthermore, our data on the confocal microscopy parallels the results on the expression of the same proteins by quantitative PCR analysis using mRNA.

  1. Additional data such as quantitative chemical evaluation and/or western blots would strengthen this manuscript.

Answer: we decided to use confocal microscopy analysis instead of western blots so that we could visualize the arrangement of the proteins in the different conditions.

  1. Some discussion of the importance of the sulfate changes observed would seem appropriate.

Answer: as requested we expanded in the discussion the changes in sulfation of the disaccharides as well as in the uronic acid units, such as beta-D-glucuronic acid and alpha-L-iduronic acid.

Reviewer 2 Report

Comments and Suggestions for Authors

In the article: “Alterations in the Structure, Composition, and Organization of Galactosaminoglycan-Containing Proteoglycans and Collagen Correspond to the Progressive Stages of Dupuytren's Disease”, the authors explored the content and organization of glycosaminoglycans, proteoglycans, and collagen in the ECM of patients at various stages of DD.

 In this  manuscript the authors  explain the rational of the study and discussed the topic point by point.

We would like to invite the authors  to better clarify some points:

1.       Please check the check punctuation and spaces;

2.       Within the introduction, the authors described GAGs, in particular HA. In this context, the following reference should be introduced; La Gatta, A. et al. (2021). Hyaluronan and Derivatives: An In Vitro Multilevel Assessment of Their Potential in Viscosupplementation. Polymers13(19), 3208. https://doi.org/10.3390/polym13193208;

3.       Within materials and methods section the authors said “These samples were categorized based on contracture severity (stages I, II, III, IV)”. Please, add more details about this categorization;

4.       Why did you use chondroitinase if they are not present in human organisms?

5.       Did you perform also tests with hyaluronidases?

6.       In your experiments (e.g fig 1 and 3) did you check the significance also between the different stage of disease?

7.       Figure 4; some images are missing of scale bar;

8.  Figure5; scale bars are not visible, maybe by changing their color it should be better;

9.  Please, always specify the magnification used for images capture;

10.   Conclusions are missing.

Comments on the Quality of English Language

In the present very minor editing errors are present

Author Response

We would like to thank the reviewers for their comments which certainly added to the manuscript in its present format.

Reviewer # 2

In this manuscript the authors explain the rational of the study and discussed the topic point by point. We would like to invite the authors to better clarify some points:

  1. Please check punctuation and spaces;

Answer: as requested we checked punctutation and spaces. Thanks for your comment.

  1. Within the introduction, the authors described GAGs, in particular HA. In this context, the following reference should be introduced; La Gatta, A. et al. (2021). Hyaluronan and Derivatives: An In Vitro Multilevel Assessment of Their Potential in Viscosupplementation. Polymers, 13(19), 3208. https://doi.org/10.3390/polym13193208;

Answer: as requested we included the reference in the manuscript is its present format.

  1. Within materials and methods section the authors said “These samples were categorized based on contracture severity (stages I, II, III, IV)”. Please, add more details about this categorization;

Answer: as requested we have included more details about the categorization.

  1. Why did you use chondroitinase if they are not present in human organisms?

Answer: we used the chondroitinases to determine the fine structure of chondroitin sulfate and dermatan sulfate. We did not use hyaluronidases since this class of enzymes does not act when we have alpha-L-iduronic acid in the structure of the galactosaminoglycan. Using chondroitinases AC and ABC we were able to clearly demonstrate the disaccharide composition of the compounds, showing the unequivocal increase in the iduronic acid units of the dermatan sulfate.

  1. Did you perform also tests with hyaluronidases?

Answer: this topic has been already answered above.

  1. In your experiments (e.g fig 1 and 3) did you check the significance also between the different stage of disease?

Answer: Regarding Figure 1, the different stages of the disease show significant differences (p<0.05). Regarding Figure 3, there is a significant difference when comparing stages I and II to stages III and IV (p<0.05) for versican expression with an increase with the progression of the disease. On the other hand, there is a decrease in the expression of decorin when comparing stages I and II to stages III and IV (p<0.05). 

  1. Figure 4; some images are missing of scale bar;

Answer: we thank the reviewer for the careful reading of the manuscript. The scale bar is now added to all images.

  1. Figure 5; scale bars are not visible, maybe by changing their color it should be better;

Answer: we thank the reviewer for the careful reading of the manuscript. The scale bar now appears in white background.

  1. Please, always specify the magnification used for images capture;

Answer: we thank the reviewer and the magnification used is now specified in the figure legends.

  1. Conclusions are missing.

Answer: we decided not to include a section related with conclusions, since the main results and conclusions are already present in the discussion section.

Comments on the Quality of English Language

In the present very minor editing errors are present

Answer: we checked the entire manuscript for errors.

Hoping to hear from you soon.

Yours sincerely,

Helena B Nader

Professor

Department of Biochemistry

Institute of Pharmacology and Molecular Biology

Round 2

Reviewer 1 Report

Comments and Suggestions for Authors

No additional comment